# Carbon-climate feedback higher when assuming Michaelis-Menten kinetics of respiration

Christian Beer[1, 2]

[1]Department of Earth System Sciences, Faculty of Mathematics, Informatics, and Natural Sciences, University of Hamburg, Hamburg, 20134, Germany
[2]Center for Earth System Research and Sustainability, University of Hamburg, Hamburg, 20134, Germany

*Correspondence to*: Christian Beer (christian.beer@uni-hamburg.de)

**Abstract.**

Earth System Models simplify complex terrestrial respiration processes assuming a first-order chemical reaction or assuming a Michaelis-Menten kinetics. The effect of the respective mathematical representation on the terrestrial carbon-climate feedback is unclear. Using a simplified model of biogeochemical feedbacks to climate, I show that the terrestrial carbon-climate feedback roughly doubles when assuming Michaelis-Menten kinetics of respiration. Consequently, the remaining carbon budget to keep global warming below 2 °C is substantially higher. The effects of the respiration formulation also depend on the underlying emission scenario. These results highlight the importance of an increased understanding of the respiration processes on a global scale for more reliably project future carbon dynamics and climate, related feedback mechanisms, and thus to estimate a valid remaining anthropogenic carbon budget using Earth System Models.

## 1 Introduction

The anthropogenic emission of carbon dioxide into the atmosphere since the industrialization period led to a global warming of about 1 K due to the greenhouse effect (Canadell et al., 2023). However, less than half of the anthropogenically emitted carbon remains in the atmosphere because terrestrial ecosystems and the ocean take up 34% and 25%, respectively (Friedlingstein et al., 2023). The main reasons for this strong carbon dioxide uptake in terrestrial ecosystems are biogeochemical feedbacks (Cox et al., 2000). Increasing atmospheric carbon dioxide ($CO_2$) concentration leads to an enhanced photosynthesis rate, and hence to a $CO_2$ uptake by vegetation on land (Cramer et al., 2001; O'sullivan et al., 2022). This carbon is stored in vegetation pools and ultimately transferred to soils by exudation, litterfall, and mortality processes, thereby increasing the soil carbon content. This is the important carbon-concentration feedback mechanism (Arneth et al., 2010) (Fig. 1) which is a negative feedback, hence responsible for the current net $CO_2$ sink on land that has been preventing us from an even stronger climate change. In contrast, autotrophic and heterotrophic respiration are also higher than under pre-industrial conditions (Canadell et al., 2023) due to i) higher substrate availability and ii) the positive temperature sensitivity of respiration (Lloyd and Taylor, 1994). This temperature sensitivity of respiration forms the positive carbon-climate feedback mechanism (Fig. 1): Higher $CO_2$ concentration leads to higher temperature, which increases respiration and hence leads to an even higher atmospheric $CO_2$ concentration (Arneth et al., 2010).

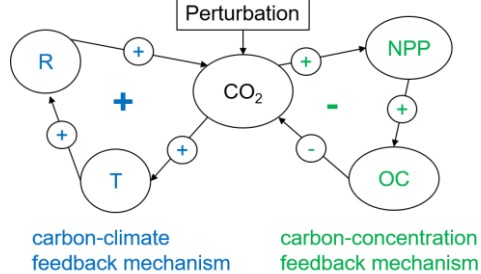

**Figure 1:** Feedback diagram for two main terrestrial biogeochemical feedback mechanisms. NPP: Net primary production, R: Respiration, OC: Land organic carbon stocks, T: Global surface air temperature, $CO_2$: Atmospheric carbon dioxide content.

These two biogeochemical feedback mechanisms have been identified as two major feedback mechanisms in the Earth system with great impact on climate (Friedlingstein et al., 2006; Arora et al., 2020). Currently, the positive carbon-climate feedback is lower than the negative carbon-concentration feedback and therefore land ecosystems act as a natural sink of $CO_2$ of about 3 Pg C per year (Friedlingstein et al., 2023). However, due to internal dynamics of the system, climate change, and changes in anthropogenic $CO_2$ emissions, the future strength of the feedback mechanisms and hence the net $CO_2$ exchange between land and atmosphere remains unclear. Recent accumulation of soil carbon in concert with higher future temperature and a declining increase in productivity can lead to a decreasing land sink under increasing $CO_2$ emissions in future (Jones et al., 2023; Cramer et al., 2001). To estimate such feedbacks, we need to run a modified version of an Earth System Model in which only one feedback mechanism is considered. The temporal difference in atmospheric $CO_2$ concentration from such experiments to model runs without the feedback is used to quantify these feedbacks (Hansen et al., 1984).

For the carbon-climate feedback mechanism (Fig. 1), the representation of respiration processes in Earth System Models is crucial. Several assumptions about the underlying processes and respective mathematical representations have been proposed. Land surface models usually represent respiration as a linear function (first-order kinetics) to the amount of available substrate (organic carbon, C),

$$\frac{dC}{dt} = -k \cdot C \qquad (1)$$

using several carbon pools (Sitch et al., 2003; Brovkin et al., 2013; Tang et al., 2022), with different decomposition rate constants k. In doing so, we assume that the active microbial biomass pool increases in relation to increased substrate availability. However, the underlying biochemical reactions are mostly enzymatic, hence a Michaelis-Menten kinetics model has been proposed to represent the dynamics of respiration (Wieder et al., 2013; Yu et al., 2020)

$$\frac{dC}{dt} = v_{max} \frac{C}{K_M + C} \qquad (2)$$

where $v_{max}$ is the maximum reaction rate under infinite carbon substrate $C$, and $K_M$ represents the amount of carbon at which the reaction rate is half of the maximum. In this model, we assume a constant active microbial biomass pool. The nonlinear shape of this relationship between reaction rate and substrate availability (in contrast to the linear dependency of first-order kinetics models) leads to a steep increase of the reaction rate under low substrate availability while only a moderate to negligible increase under high substrate availability. In doing so, this model implicitly represents the function of enzymes in the underlying biochemical reactions. It has been proposed, that such model enables a more valid aggregation from the process level (e.g. rhizosphere, aggregatusphere) to the landscape scale (Reichstein and Beer, 2008). However, both models are great simplifications of the underlying biogeochemical processes with strong assumptions. Therefore, in this study I will

use both equations to represent respiration and study the resulting structural uncertainty in feedbacks and remaining
C budgets.
The two approaches represented by equations 1 and 2 imply different responses of respiration to changing substrate
availability. Therefore, future dynamics of respiration should differ depending on the mathematical formulation.
Such structural model uncertainty is in particular of interest because there might be a point when the land sink
starts to decrease even under continuing high anthropogenic emissions(Cramer et al., 2001), or for the question on
how land sinks will react to decreasing or even negative anthropogenic carbon emissions.
Therefore, I ask three main questions in this paper: What is the effect of the respiration model structure on
- projections of the land carbon sink
- the strength of the carbon-climate feedback
- the remaining anthropogenic carbon budget
under different carbon emission scenarios? To address these questions I performed a feedback analysis using a
simplified but process-based model of global biogeochemical feedback mechanisms twice, using a first-order and
a Michaelis-Menten kinetics model of respiration. The simplified model of global biogeochemical feedback
mechanisms is of zero dimension (only globally aggregated pools) and neglects many detailed processes and
interactions between ecosystem components. Therefore, the idea is not to precisely quantify C budgets or feedback
but rather show qualitatively the effects of the respiration model structure on these estimates.
**2 Methods**
**2.1 Simplified Carbon-Climate Feedback Model**
The model has been designed to study the two major biogeochemical feedbacks to climate displayed in Fig. 1.
Exchanges of carbon among atmosphere, ocean and land are represented using a reduced number of carbon pools
without spatial details but still in a process-based way, i.e. based on a set of differential equations. For example,
the amount of carbon taken up by vegetation depends on the atmospheric carbon content while the amount of $CO_2$
released to the atmosphere due to respiration depends on the carbon content of the ecosystem. The model assumes
a global surface air temperature response to changing atmospheric carbon dioxide content using a transient climate
response parameter, which is lagged due to the ocean heat capacity. The model is driven by anthropogenic carbon
dioxide emissions to the atmosphere following several scenarios developed for the IPCC 6th assessment report.
A detailed description of the model can be found in (Lade et al., 2018). Here, I apply two alternative model
versions, one assuming a first-order kinetics of respiration (FOK), and one assuming a Michaelis-Menten kinetics
of respiration (MMK). The representation of terrestrial carbon uptake by gross primary productivity is identical in
both model versions. It is assumed to increase logarithmically with atmospheric carbon dioxide $C_a$ (Equations 3
and 4, first term right-hand side). In addition, emissions due to land use change $E_L$ are subtracted the same way in
both versions, and the increase in respiration with temperature is represented by a typical $Q_{10}$ model (Equations 3
and 4, second term right-hand side). Only the dependence of respiration to land carbon stocks differs. The FOK
model assumes a first-order kinetics with a respiration rate constant estimated by pre-industrial GPP and carbon
stocks, $k = \frac{GPP_0}{C_{L,0}}$ following the same principle as in (Lade et al., 2018).
$\frac{dC_L}{dt} = GPP_0 \left(1 + \alpha \log \frac{C_a}{C_{a,0}}\right) - Q^{\frac{\Delta T}{10}} \cdot k \cdot C_L - E_L$     (3)
In contrast, the MMK model represents respiration as a classical Michaelis-Menten equation with parameters $v_{max}$
and $K_M$:

$$\frac{dC_L}{dt} = GPP_0 \left(1 + \alpha \log \frac{C_a}{C_{a,0}}\right) - Q^{\frac{\Delta T}{10}} \cdot v_{max} \frac{C_L}{K_M + C_L} - E_L \qquad (4)$$

Parameters and pre-industrial pools and fluxes for model initialization were taken from (Lade et al., 2018) and
partly adjusted (Table 1). The transient climate response to $CO_2$ doubling $\lambda$ is set at the higher end of the range
reported for CMIP6 model results (Arora et al., 2020) in order to match the observed historical temperature
anomaly. Parameters $v_{max}$ and $K_M$ of Equation 4 are optimized using a standard gradient decent approach
(MATLAB R 2023b function lsqnonlin) such that the difference of the modelled and observation-based land
carbon changes is minimized.
**Table 1.** Value and description of parameters different from Lade et al. (2018).

| Name | Symbol | Value | Reference/comment |
|---|---|---|---|
| Pre-industrial soil and vegetation Carbon | $C_{L,0}$ | 2305 Pg C | Sum of vegetation and soils carbon stocks following Canadell et al. (2023), and C stocks of the active layer of gelisols following Hugelius et al. (2014) |
| Transient climate response to $CO_2$ doubling (TCR) | $\lambda$ (Equation 10 of (Lade et al., 2018)) | 2.5 K | Tuning parameter, higher end of range of CMIP6 models (Arora et al., 2020; Nijsse et al., 2020) |
| Respiration sensitivity parameter | $Q$ | 2 | Vaughn and Torn (2019) |
| Pre-industrial GPP | $GPP_0$ | 113 Pg C a$^{-1}$ | Friedlingstein et al. (2023) |
| $CO_2$ sensitivity of GPP | $\alpha$ | 0.35 | Tuning parameter (Alexandrov et al., 2003) |
| Max respiration rate in MMK model | $v_{max}$ | 200 Pg C a$^{-1}$ | Tuning parameter |
| Substrate concentration at half of max respiration rate in MMK model | $K_M$ | 1787 Pg C | Tuning parameter |


## 2.2 Modelling protocol

The two model versions have been run from 1850 until 2100 using a daily time step forced by anthropogenic
carbon dioxide emissions from fossil fuel burning and from land-use change. For this, I combined reported
historical emissions from the Global Carbon Project (Friedlingstein et al., 2023) with Shared Socioeconomic
Pathways (SSP) emission scenarios from the public database of the Institute for Applied Systems Analysis (Riahi
et al., 2017). I selected four widely used scenarios produced for the CMIP6 protocol (Gidden et al., 2019): SSP1-
26 (optimistic scenario, reaching economic growth while retaining sustainability and reducing inequalities), SSP2-
45 (including mitigation strategies), SSP3-70 (represents a future of inequality and fossil fuel dependency), and
SSP5-85 (representing economic growth through strong reliance on fossil fuels). These scenarios reach a forcing
of 2.6, 4.5, 7.0, and 8.5 W/m² at the end of the century and represent a huge spread of carbon emissions into the
atmosphere (Fig. 2). I interpolated linearly the reported emissions at decadal scale to an annual resolution. In the
combined time series (Fig. 2), historical emissions span the period 1850-2014 and scenarios continue from 2015
until 2100.

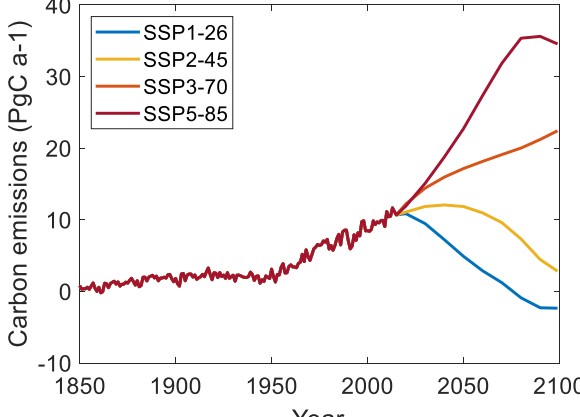


**Figure 2:** Total $CO_2$ emissions from burning fossil fuels and land-use change from combining a historical dataset with results
from Integrated Assessment Models for different scenarios.

I performed model simulations for these emission scenarios and for both model versions, FOK and MMK. The
results were used to evaluate the model during the historical period, and to estimate the remaining carbon budgets
to keep warming below a certain threshold. For the feedback analysis, all these simulations were repeated three
times. To estimate the feedback factor, I did model simulations in which only the terrestrial carbon-climate
feedback is considered. The results were used to estimate the respective $\Delta C_A^{on}$ (section 2.4). For calculating the
feedback sensitivities β and γ (Section 2.4), I additionally performed biogeochemically and radiatively coupled
simulations following (Friedlingstein et al., 2006; Lade et al., 2018) and derived $\Delta C_L$, $\Delta C_A$ and $\Delta T$ from these
simulations. In the biogeochemically coupled simulation, I set λ to 0, hence effects of $CO_2$ change on temperature
are excluded. In the radiatively coupled simulation, I neglected all effects of $CO_2$ on terrestrial or marine carbon
pools. In total, that are 32 model simulations.

**2.4 Feedback analysis**
Atmospheric carbon content increases in time due to annual anthropogenic emissions ($e_i$) and internal feedback
mechanisms. To estimate this carbon dioxide change when considering a terrestrial carbon-climate feedback
("on"), I averaged the atmospheric carbon content during a reference period in the future (2080-2100) and in the
past (1850-1900) using the respective model simulation (section 2.3), and subtract both:
$\Delta C_A^{on} = C_A^{future} - C_A^{past}$  (5).
The respective atmospheric carbon change without considering the feedback ("off") equals the sum of emissions:
$\Delta C_A^{off} = \sum_{i=1850}^{2100} e_i$      (6).
The feedback is the difference $\Delta C_A^{on} - \Delta C_A^{off}$ and the feedback factor F is the ratio of both changes, which can be
used to compare feedbacks and to identify positive (F>1) or negative feedbacks (F<1) (Cox et al., 2000;
Friedlingstein et al., 2003; Hansen et al., 1984; Zickfeld et al., 2011)
$F = \frac{\Delta C_A^{on}}{\Delta C_A^{off}}$      (7).
Sensitivities of the land carbon change to atmospheric carbon concentration (β) and temperature changes (γ) are
defined following (Friedlingstein et al., 2006; Heinze et al., 2019) as
$\Delta C_L = \beta \cdot \Delta C_A + \gamma \cdot \Delta T$      (8).
I used the biogeochemically coupled simulation results to estimate β (ΔT=0), and the radiatively coupled results
to estimate γ (ΔC$_A$=0).
**3 Results**
Model results of carbon fluxes and the surface temperature anomaly for the historical period are in general
agreement with results by the Global Carbon Project (Friedlingstein et al., 2023) and the NOAA Global Surface
Temperature record (Fig. 3), i.e. the overall historical trends are captured. The model does not represent spatial
details, oversimplifies functional diversity and does not represent certain processes, such as disturbances.
Therefore, the model is not able to capture the inter-annual variability of land carbon fluxes (Fig. 3). The general
long-term agreement shows that major biogeochemical feedback mechanisms are correctly represented, and that
initial conditions (Table 1) and model parameters (Table 1) are reasonable. Therefore, we assume that we can
apply this model to study the effects of structural respiration model uncertainty on the carbon-climate feedback
strength.

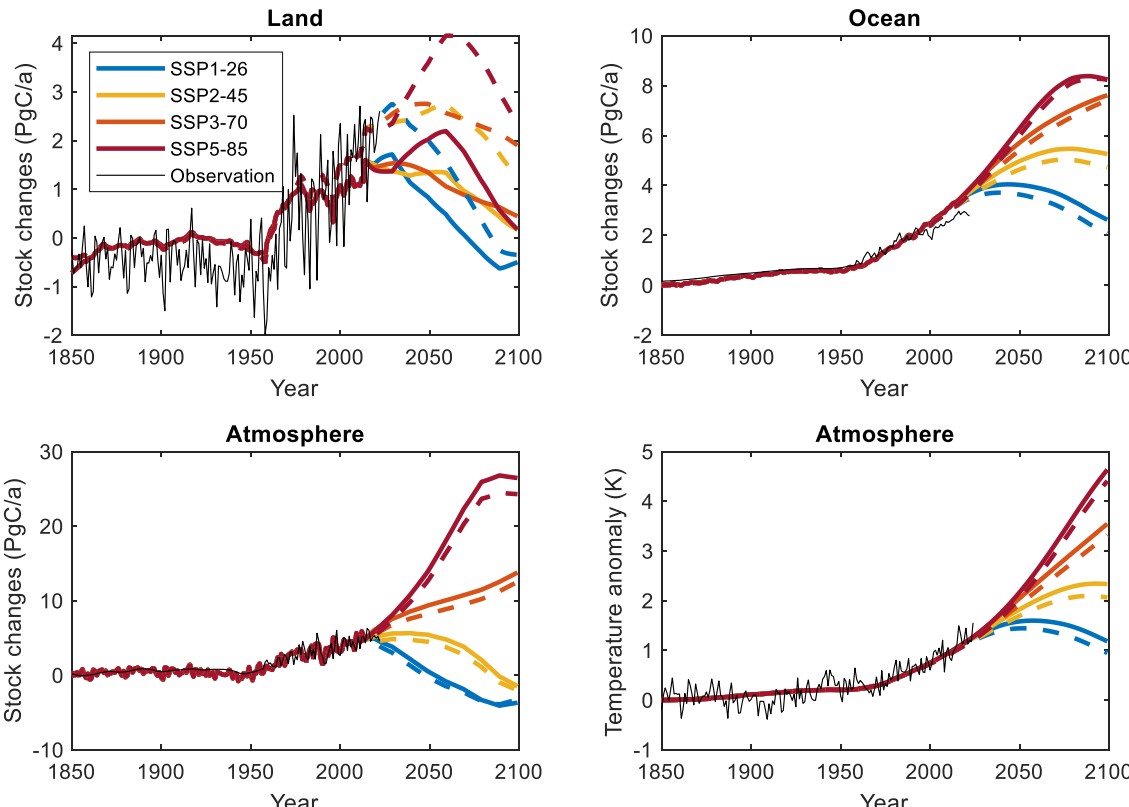

**Figure 3:** Simulated carbon fluxes and temperature anomaly for the different scenarios. FOK and MMK model results are displayed by solid and dashed lines, respectively. Simulation results are compared to estimates by the Global Carbon Project or to the NOAA Global Surface Temperature record, which has been bias corrected to the model results to match reference periods.

Figure 3 also shows the projections of carbon fluxes to land, ocean and atmosphere, as well as the temperature change for the two different model structures until 2100 following the different emission scenarios. Overall, these projections of the main carbon cycle fluxes and temperature change are similar to concentration-driven CMIP6 results (Canadell et al., 2023; Jones et al., 2023). The projected ocean carbon sink in this study is substantially higher for most of the scenarios, and the land carbon sink of the model using a first-order kinetics respiration approach (FOK) is lower, but comparable with Lade et al. (2018). Otherwise, the projections of the change in atmospheric carbon stocks and the global surface temperature change are similar to studies using Earth System Models (Canadell et al., 2023). However, spread of carbon cycle projections using other models is usually also very high (Canadell et al., 2023; Jones et al., 2023), and the uncertainty due to parameter values or initial conditions hardly quantified in these studies.

The projected land sink evolution differs depending on both, the emission scenario and the model structure applied. Under high emission scenarios, the land sink continues to rise and peaks in the middle of the century followed by a decreasing sink until 2100. This peak has been reported by DGVMs and ESMs before (Cramer et al., 2001; Jones et al., 2023) and is due to the reverse shape of the two main response functions, logarithmic productivity response to elevated $CO_2$ and quasi-exponential respiration response to temperature. A second reason is internal carbon dynamics: Respiration depends on the amount of land carbon stocks, which continued to increase until some maximum and therefore is the basis for a high respiration flux during the following time. For the scenario SSP1-

26, the land sink starts to decrease immediately after the historical period, i.e. when emissions are reduced, and
depending on the model structure is getting even negative at the second half of the century.
The projected land carbon sink in 2100 is much higher when assuming a Michaelis-Menten kinetics model for
respiration (MMK) even under an equal temperature sensitivity of respiration as by the first-order kinetics model
(FOK), and even when parameters are chosen to fit both model results during the historical period. In addition, the
peak in the middle of the century is more pronounced when using the MMK model (Fig. 3). Hence, this difference
is only due to internal carbon dynamics differences, in particular a non-linear (decreasing) change of the respiration
rate with increasing substrate availability under when assuming Michaelis-Menten kinetics. This clearly
demonstrates the uncertainty of land carbon sink dynamics just due to alternative assumptions and mathematical
formulations of respiration processes. As a result of higher land sinks using the MMK model, ocean and
atmosphere sinks are smaller and the temperature change is lower (Fig. 3). Due to the higher land C sink assuming
Michaelis-Menten kinetics, also total changes in land carbon stocks are much higher, i.e. land takes up several
hundred of Pg C more depending on the emission scenario.

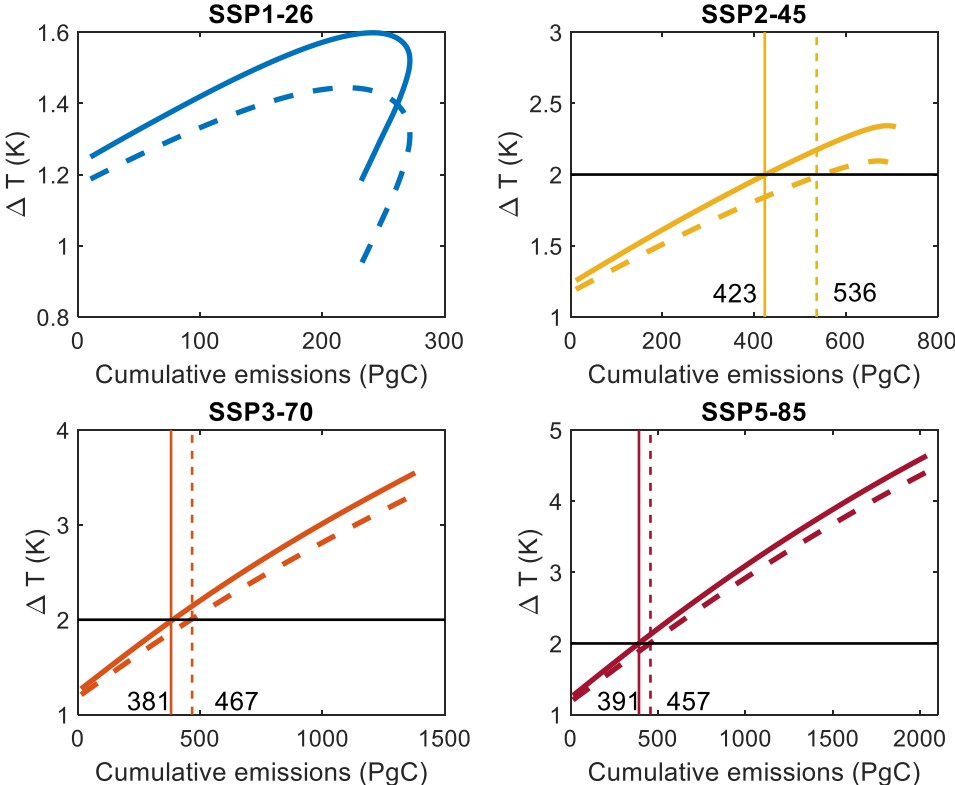


**Figure 4**. Relationship between global air surface temperature difference to pre-industrial temperature and the cumulative
emission of $CO_2$ from 2024 until 2099 for different emission scenarios and the two model simulations FOK (solid lines) and
MMK (dashed lines). Horizontal lines indicate a temperature change threshold of 2 K, and vertical lines and numbers indicate
the respective cumulative emissions since 2024 to reach that temperature change target.

These differences in the projected land sinks do have clear consequences for the Transient Climate Response to
Cumulative Emissions of Carbon Dioxide (TCRE) and hence the remaining anthropogenic carbon budgets under
different emission scenarios. Usually, there is a quasi-linear relationship between the cumulative emission and the
temperature change (Fig. 4). Under reduced emissions of SSP1-26 scenario, ocean and land C uptake may remain
high (blue curves in Fig. 3), leading to a hysteresis in the TCRE (Koven et al., 2023). Such hysteresis is not visible
in the other scenarios (Fig. 4) because emission reductions are not strong enough (Fig. 2). Interestingly, the

relationship is less steep and more non-linear for the MMK model for all scenarios. From the TCRE the remaining carbon budget for a certain temperature threshold can be estimated (Canadell et al., 2023). In Fig. 4, the vertical lines indicate the amount of emissions since 2024 that - according to this model - can be still emitted in order to keep warming below the threshold of 2 °C warming compared to the pre-industrial situation, which is indicated by the horizontal line. We skip this analysis for scenario SSP1-26 results because the MMK model fails to reach a 2 °C increase at all (Fig. 4). For the other emission scenarios, the FOK model suggests 381 to 423 Pg C that can be emitted to the atmosphere in order to keep warming below 2 °C compared to pre-industrial temperature (Fig. 4). These estimates are slightly higher than the median remaining C budget estimated by CMIP6 experiments using ESMs of 370 Pg C (table 5.8, (Canadell et al., 2023)). Importantly, when assuming a Michaelis-Menten kinetics of respiration (MMK), the remaining C budget is higher and range between 457-536 Pg C. This is due to flatter slopes of these model results (Fig. 4).

**Table 2.** Terrestrial carbon-climate feedback (Pg C) for different representations of respiration in the model. Shown is the difference of model results accounting for the feedback and excluding it based on the temporal change in atmospheric carbon content between 2080-2100 and 1850-1900.

|  | First-order kinetics (FOK) | Michaelis-Menten kinetics (MMK) |
|---|---|---|
| SSP1-26 | 379 | 920 |
| SSP2-45 | 423 | 955 |
| SSP3-70 | 431 | 959 |
| SSP5-85 | 498 | 1019 |

Using the first-order kinetics approach of respiration (FOK), I estimate a carbon-climate feedback of 379 to 498 Pg C when comparing the average $CO_2$ concentration of the period 2080-2100 with pre-industrial conditions, depending on the emission scenario (Table 2). This translates into feedback factors of 1.2 to 1.4 (Table 3), which are similar to previous estimates (Lade et al., 2018). Interestingly, the strength of the feedback mechanism as expressed by the feedback factor decreases with increasing carbon emissions (Table 3), i.e. the internal Earth system interactions are more important under reduced anthropogenic emissions. However, when assuming Michaelis-Menten kinetics of respiration, this carbon-climate feedback strength is higher (Table 3) depending on the underlying scenario.

**Table 3**. Feedback factor of the terrestrial carbon-climate feedback for different representations of respiration in the model. Shown is the difference of model results accounting for the feedback and excluding it based on the temporal change in atmospheric carbon content between 2080-2100 and 1850-1900.

|  | First-order kinetics (FOK) | Michaelis-Menten kinetics (MMK) |
|---|---|---|
| SSP1-26 | 1.41 | 2.0 |
| SSP2-45 | 1.31 | 1.71 |
| SSP3-70 | 1.23 | 1.52 |

| SSP5-85 | 1.21 | 1.43 |
|---------|------|------|

The FOK model estimates the sensitivity of the land carbon change to increasing atmospheric CO2 concentration (β, Table 4) to be 1.4 Pg C ppm$^{-1}$ assuming the high-emission scenario SSP5-8.5. This is similar to CMIP4 model runs using the high-emission scenario SREAS A2 (Friedlingstein et al., 2006) and at the higher end of the range of CMIP6 model results without considering the N cycle in 4xCO2 experiments (Arora et al., 2020). Interestingly, the sensitivity increases towards scenarios assuming less emissions (Table 4), and the sensitivity is higher when assuming a Michaelis-Menten kinetics of respiration (Table 4). The land carbon change sensitivity to climate change (γ, Table 4) is estimated at -117 Pg C K$^{-1}$ in this case. This is at the higher end of the range for the previously mentioned ESM results (Arora et al., 2020; Friedlingstein et al., 2006). This parameter is also more negative when assuming Michaelis-Menten kinetics or when considering a lower emission scenario (Table 4).

**Table 4**. Sensitivities of the land carbon change to changing atmospheric carbon dioxide (β , Pg C ppm$^{-1}$) and temperature (γ , Pg C K$^{-1}$) for different representations of respiration in the model (FOK and MMK).

|  | β, Pg C ppm$^{-1}$, first-order kinetics (FOK) | β, Pg C ppm$^{-1}$, Michaelis-Menten kinetics (MMK) | γ, Pg C K$^{-1}$, first-order kinetics (FOK) | γ, Pg C K$^{-1}$, Michaelis-Menten kinetics (MMK) |
|--------|--------|--------|--------|--------|
| SSP1-26 | 3.4 | 9.8 | -133 | -218 |
| SSP2-45 | 2.3 | 5.4 | -125 | -204 |
| SSP3-70 | 1.6 | 3.4 | -124 | -198 |
| SSP5-85 | 1.4 | 2.7 | -117 | -187 |

## 4 Discussion

Besides gross primary productivity, ecosystem respiration is one of the main land-atmosphere carbon exchange processes (Friedlingstein et al., 2023). The underlying biochemical processes are complex and mathematical models of simplified net reactions are usually applied in Earth System Models: Either assuming a first-order chemical reaction of carbon and oxygen to carbon dioxide and applying Equation 1, or considering the underlying enzymatic reactions and hence applying Equation 2. The epistemic uncertainty in projecting future land-atmosphere exchange of $CO_2$, climate and the related biogeochemical feedbacks underlying these assumptions have been addressed in this paper. Model parameters have been chosen based on literature values, and to fit published historical carbon and temperature changes (section 2.3) for the first-order kinetics approach (FOK). For the Michaelis Menten kinetics model (MMK), we selected parameter values such that results are also similar to Global Carbon Budget estimates and the FOK model during the pre-industrial period. Interestingly, effects of anthropogenic carbon emissions on future land sink dynamics differ between both model versions, with several Pg C per year higher uptake by land when assuming Michaelis-Menten kinetics for respiration (Fig. 3). Such higher land carbon uptake leads to a lower ocean carbon sink hence increasing differences between land and ocean sinks. In addition, the projected global surface temperature change until 2100 is lower in the MMK model (Fig. 3), i.e. a lower temperature change response to cumulative carbon emissions (Fig. 4). Since increasing surface temperature

will lead to an additional $CO_2$ release from land to the atmosphere, there is the positive carbon-climate feedback
mechanism (Arneth et al., 2010), and here I asked the question, is there also an effect of the respiration model
structure on this feedback strength?
Indeed, this feedback roughly doubles when assuming Michaelis-Menten kinetics, and it is higher for strong carbon
emission scenarios (Table 2). As a consequence, the model results imply a higher remaining anthropogenic carbon
budget to keep warming below 2 °C above pre-industrial levels of up to circa 100 Pg C but depending on the
emission scenario only because we assume an alternative model structure for respiration. These estimates are
similar to estimates of additional warming-induced C loss from permafrost-affected soils until 2100 of 10-100 Pg
C (Koven et al., 2015). Other additional Earth system feedbacks currently not represented in Earth System Models
(section 5.5.2.2.5 in Canadell et al. (2023)), and additional geophysical uncertainties like non- $CO_2$ forcing or
emission uncertainty (Table 5.8 in Canadell et al. (2023)) are also of the same order of magnitude. The structural
uncertainty in the formulation of respiration is also of the same order of magnitude as the total annual gross primary
production or the respiration flux (both 130 Pg C yr-1, Friedlingstein et al. (2023)). Shall we assume a linear or
non-linear dependence of respiration on the amount of substrate? This assumption influences the internal land
carbon dynamics, because in the latter case respiration does not respond to higher substrate availability in the same
way as in the linear model. This is also visible when looking at the sensitivities of the land carbon change to $CO_2$
change (β, Table 4) which roughly double when assuming Michaelis-Menten kinetics because the response of
respiration to higher substrate availability is lower.
I applied a simplified model of global biogeochemical feedback mechanisms, considering only one terrestrial
carbon pool, and hence integrating autotrophic and heterotrophic respiration,and no explicit pool of microbial
biomass and microbial functions. Therefore, many specific underlying processes and interactions of ecosystem
components are neglected. For example, an increase in heterotrophic respiration due to increasing plant
productivity and carbon input to soils (priming effect, (Fontaine et al., 2007; Keuper et al., 2020)), or changing
microbial community structure as a response to climate change (Glassman et al., 2018) is not considered. Nutrient
limitation of vegetation productivity (Hungate et al., 2003) is only implicitly parametrized in Equations 3 and 4
through a logarithmic response function of GPP to $CO_2$. Hence, I do not quantify the effects of nutrient availability
on the carbon-climate feedback in addition to the effects of either respiration model used. When assuming a MMK
model, increasing $CO_2$ leads to higher increase in land C stocks (β, Table 4) due to lower respiration. However,
this mechanism can, for instance, also lock more nutrients in soil organic matter hence change the response
function of GPP to $CO_2$. When considering nutrient processes, land C change sensitivities to $CO_2$ and temperature
have been shown to be much smaller (Arora et al., 2020). In addition, climate change is expressed as a temperature
change in this model and precipitation effects on carbon cycle functions (Jung et al., 2017) are not taken into
account. Therefore, the presented results are first conservative estimates which should be verified using a state-of-
the-art ESM including nutrient cycles and Michaelis-Menten kinetics (Yu et al., 2020).
Besides structural uncertainty, an additional relevant source of uncertainty of such highly parametrized model is
parameter uncertainty. Interestingly, the additional analysis presented in the supplementary material shows that
the structural uncertainty of the carbon-climate feedback due to the respiration equation is higher than the
parameter uncertainty, regardless of the emission scenario applied.
Still, the presented results point to the importance to communicate and address existing structural uncertainties in
Earth System Models. Just by assuming an underlying Michaelis-Menten kinetics of respiration processes leads to
distinct projections of future respiration and the carbon-climate feedback mechanism. These results also
demonstrate the need for novel research clarifying a valid process-based model structure of ecosystem respiration.

**5 Conclusions**

Two major gross carbon fluxes govern the recent land carbon sink, photosynthesis and respiration. While detailed
process-based photosynthesis models have been developed and applied in Earth System Models, how to model
respiration processes remains unclear. The model structure of respiration alone can lead to a doubling of the
carbon-climate feedback estimate over the 21$^{st}$ century. Depending on the underlying emission scenario, that
translates into a substantial difference of the remaining carbon budget to keep global warming below 2 °C of up
to circa 100 Pg C depending on the emission scenario. These results show the importance of an increased
understanding of the mathematical model structure of respiration processes in Earth System Models for more
reliably projecting future carbon dynamics and climate, related feedback mechanisms, and hence to estimate a
valid remaining anthropogenic carbon budget.

**Code availability**

MATLAB code of the model versions applied is available via zenodo at https://doi.org/10.5281/zenodo.15696851.

**Data availability**

All required data to run the model and reproduce the results is available online and open access.
**Author contribution** CB designed the study, wrote the computer code, downloaded and interpolated the emission
data, run the models, analysed the results and wrote the manuscript.
**Competing interests** CB declares that he has no conflict of interest.
**Acknowledgements** CB acknowledges financial support by Deutsche Forschungsgemeinschaft through the
Heisenberg program (508047523).

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
