# Peer review of "Carbon-climate feedback higher when assuming Michaelis-Menten kinetics of respiration"

_EGUsphere, 2024_

## Referee Comment (RC2)

**Review on**

**Christian Beer, "Carbon-climate feedback higher when assuming Michaelis-Menten kinetics of respiration"**

**submitted to Earth System Dynamics**

August 8, 2024

It is well known that even at the conceptual level there are large uncertainties in the representation of the global carbon cycle and its coupling with climate (Canadell et al., 2023, section 5.7, p. 769)[1]. To improve the reliabilty of coupled climate-carbon simulations, in particular the modelling of soil respiration is an active field of research [8, 7]. The study reviewed here contributes to this field by performing scenario simulations with a zero dimensional climate-carbon model (Lade et al., 2018) to investigate the effect of a modified representation of soil respiration on the future land carbon sink, the size of the carbon-climate feedback, and the "remaining carbon budget", i.e. the amount of future anthropogenic emissions that will not compromise the 2 degree climate warming target. Here the original representation of soil respiration by a first order kinetics (FOK) is replaced by a Michaelis-Menten (MMK) type representation.

I agree to reviewer #1 that, qualitatively, the results from this study are rather foreseeable: Because the MMK model limits soil respiration when the land carbon storage grows beyond a certain amount, this model enhances the land carbon sink, whereby the remaining carbon budget gets larger, the land contribution to the carbon-concentration feedback gets more positive (there is more carbon stored per ppm $CO_2$ rise; $\beta$ sensitivity), and the land contribution to the climate-carbon feedback gets more negative (there is more soil carbon to be respired per degree climate change; $\gamma$ sensitivity). So the main advancement by this study would be its quantitative results (that are partly noted in the abstract). But I doubt that those numbers have any relevance (major comment 1 below) and, moreover, I don't trust these numbers (major comment 2 below). In addition, as also noted by reviewer #1, the feedback analysis is rather unclear (major comment 3 below), and – if my reconstruction of its meaning is right – the feedback metrics used are not simply a measure of the carbon-climate feedback (as intended by the author; see title) but are mixing information on this and the carbon-concentration feedback.

**Major comments**

**1.** I fear that the quantitative results of this study are, if at all, only of academic interest. By the Michaelis-Menten type formula used by the author, soil respiration starts to saturate at large land carbon storage. The author motivates the usage of this model because "the underlying biochemical reactions are mostly enzymatic" (line 57), a rather incomplete justification. In fact, such a saturation appears because at large carbon storage also huge amounts of nutrients are bound in the organic soil molecules that are thus not available for the growth and maintenance of the microbials that mineralize them. These mineralized nutrients are at the same time necessary for the plants binding those nutrients to flourish. Accordingly, the growth of plants and the respiration of soils is
* * *
[1]Only references additional to those in the reviewed paper are listed below.

closely linked (see e.g. [9, chapter 10], [4]) so that the input to the land carbon stocks by photosynthesis (net primary productivity) is via the nutrient connex highly correlated with the loss of land carbon by soil respiration. This link, necessary to obtain realistic estimates of the feedbacks and remaining carbon budgets, is missing in the study reviewed here because only limitation to soil respiration is considered, but not a parallel limitation by plant productivity.

Accounting for this link leads to very different results: To account for nutrient limitation, the Earth system modelling community recently put a lot of effort in accounting for at least one such nutrient by complementing the carbon cycle dynamics with a description of the nitrogen cycle: 6 out of 11 Earth system models (ESMs) participating in CMIP6 were equipped with a nitrogen cycle (Canadell et al., 2023, p. 730). In a coordinated community effort, for the whole model ensemble the feedbacks (Canadell et al., 2023, section 5.4.5.5, p. 735) and remaining carbon budget (Canadell et al., 2023, section 5.5, p. 742) were analyzed, including a separate analysis of the effect of nitrogen limitation on the feedbacks (Arora et al., 2020, Fig. 5). It turns out that tendentially for the land component both the carbon-climate feedback ($\gamma$) and the carbon-concentration feedback ($\beta$) are reduced by accounting for nitrogen imitation, just the opposite of the study reviewed here (see also [10]). – Concerning the remaining carbon budget I just realized that a more comprehensive analysis of the effect of nutrient limitation in a particular ESM is currently under review in Biogeosciences [6].

**2.** Even if one ignores the previous comment, and insists to study the effect of limitation of soil respiration alone, I doubt that the simulation results can be trusted. Firstly, the author has without explanation modified the employed model by Lade et al. (2018) in a – as I think – unreasonable way: It belongs to basic understanding of the land carbon cycle that its long term dynamics is determined by the allocation into long-term stable organic componds, i.e. by net primary production (NPP), instead of gross primary production (GPP) that includes allocation to sugars that are fastly respired (see e.g. [5]). Accordingly, in the original model by Lade et al. (2018), NPP appears as input to the land carbon cycle (Lade et al., 2018, Eq. (2), but in the reviewed study the author has replaced it by GPP (Eq. (3), line 101). This replacement would be OK if in addition the model equation had been complemented by a term for carbon loss by autotrophic respiration, but this is not the case. Hence the carbon inputs to the model are overestimated by a factor two ($GPP_0$=113 PgC/a (table 1, line 144) vs. $NPP_0$=55 PgC/a (Lade et al., 2018, table 1). Together with the modified value for the pre-industrial land carbon, this modification results in a characteristic land carbon turnover time of $C_{L,0}/GPP_0 \approx 20$ years, while the respective turnover time in the original setup of Lade et al. is $C_{L,0}/NPP_0 \approx 34$ years – this is a serious modification of the dynamics of the model affecting how the carbon cycle reacts to the scenario forcing in the simulations, modifying in particular the distribution of carbon between the three compartments (land, ocean, atmosphere) in the scenario simulations and therefore also affecting the size of the feedbacks.

And secondly, the re-tuning of the model for the MMK model variant (the case with

limitation of soil respitation) is rather unclear: The author notes that the parameter $k$ of the FOK soil respiration term is obtained "following the same principle as in (Lade et al., 2018)" by setting pre-industrial soil respiration equal to pre-industrial plant productivity (lines 99-100) (but using GPP instead of NPP; see previous remark). While this is an adaptation to *pre-industrial* conditions, lateron it is remarked that "for the first-order kinetics approach (FOK)" (line 246) "model parameters [i.e. $k$] have been chosen to fit *historical* carbon and temperature changes . . ." (line 245; emphasizing by me) – I guess the latter is wrong. But more importantly: Concerning the two parameters $\nu_{max}$ and $K_M$ of the MMK soil respiration term it is remarked that they "are set such that MMK model results match FOK model results for the *pre-industrial* period" (lines 141-142; emphasizing by me). If this means that one sets following (Lade et al., 2018) as in the FOK case pre-industrial soil respiration equal to pre-industrial plant productivity, one had only one condition for two parameters, so that the problem to find $\nu_{max}$ and $K_M$ is under-determined, meaning that the author's choice of them contains a subjective decision. Hence a clear explanation how the author comes to values of $\nu_{max}$ and $K_M$ is missing. Particularly the value of $K_M$ is crucial for the whole study as it determines at what value of the land carbon stock the limitation of soil respiration sets in, so that all results of the study depend critically on the choice of this value. I personally doubt that even if one uses besides pre-industrial also the histoical data, the values of the two parameters are sufficiently constrained to come to quantitatively reliable simulation results. This point of parameter estimation is so important for this study, that it needs a clear description of how it is done and a separate analysis how uncertainties in parameter estimation affect the study results.

**3.** Already reviewer #1 has tried to get clearer about the presented feedback analysis. If I understand it correctly, the feedback analysis is performed as follows. For each model variant (FOK, MMK) a simulation is performed with the fully coupled model (simulation 'on') and another one where climate change (represented by temperature change) is not affecting the soil respiration (simulation 'off'), realized by setting $Q_{10} = 1$. According to line 127 and the captions of tables 1 and 2 the feedback analysis is based on changes in *atmospheric* carbon stocks, which is a bit surprising as the the direct effect of the modified soil respiration is on the *land* carbon stock (as naturally assumed by reviewer #1). Calling the changes in atmospheric carbon in the two simulations by $\Delta C_A^{on}$ and $\Delta C_A^{off}$, the strength of the feedback is then quantified in two ways, namely, as explained in line 127, by calculating the difference $D := \Delta C_A^{on} - \Delta C_A^{off}$ (listed in table 2) and by calculating the quotient $F := \Delta C_A^{on}/\Delta C_A^{off}$ (listed in table 3), called, following [2] and (Zickfeld et al., 2011), "feedback factor" in the reviewed study. In the 'off' simulation it is the land carbon-climate feedback that is switched off so that $\Delta C_A^{off}$ is the land contribution from the carbon-concentration feedback, meaning that $D$ quantifies the effect from the carbon-climate feedback happening *on top of the carbon-concentration feedback*. And $F$ is the factor by which carbon-climate feedback *boosts the carbon-concentration feedback*. Hence both metrics depend on the size of the carbon-concentration feedback as a reference and thus they are not a quantification of the carbon-climate feedback alone,

as claimed by the author (title, abstract, captions of tables 1 and 2, conclusions). In addition, this reference is different for the two simulated model variants (FOK, MOK) so that the comparability of in particular the dimensional feedback metric $D$ is questionable ($F$ is non-dimensional).

In summary, if my interpretation is correct, the feedback metrics used are not (as intended by the author) independent measures of the carbon-climate feedback, but are of more complex nature. As already remarked by reviewer #1, this feedback could be more directly measured by the commonly used $\gamma$ sensitivity (see e.g. (Arora et al., 2020)). And because the soil respiration affects the carbon-concentration feedback much more directly than the carbon-climate feedback, it would be natural to quantify it as well, which could be easily done by calculation of the commonly used $\beta$ sensitivity (see e.g. (Arora et al., 2020)), which may be immediately obtained from the 'off' simulation.

**Minor comments**

- Title, abstract, and throughout: Whenever the author talks of respiration, *heterotrophic* respiration is meant (also called "soil respiration"), in contrast to autotrophic rspiration. This should be made clear in the title, the abstract, and elsewhere.
- The Lade et al. model is missing the temperature dependence of photosynthetic production. I don't think that a quantification of the carbon-climate feedback without that temperature dependence makes sense. At least it should be made clear that the study accounts (if at all) only for part of that feedback.
- Line 9 (abstract): The formulation "The epistemic uncertainty . . . is unclear" a bit weird: I guess the author wants to express that because of epistemic uncertainty there is quantitative uncertainty whose size is unclear – hence it would be the quantitative uncertainty being unclear, not the epistemic uncertainty.
- Lines 13-15 (abstract): This formulation is misleading: taking it literally, the author suggests to improve our understanding of the model structure of Earth system models. But I guess the author wants to emphasize that we need to improve our understanding of heterotrophic respiration to improve its representation in Earth system models.
- Line 28: Why "in contrast"? In contrast to what?
- Line 35 (caption of Fig. 1) and line 39: Only the negative feedback may be termed "biogeochemical", the positive feedback is non-biological and mostly determined by physics (radiation in the atmosphere, temperature dependenc of reaction rates); for terminolgy see e.g. [3].
- Line 50: A better reference than (Zickfeld et al., 2011) on the methodology of separating climate-carbon feedbacks would be the original study [2] or the recent review [3].
- Line 85: Better: "atmospheric $CO_2$ concentration" instead of "atmospheric carbon content".
- Line 106: Should this read "1850" instaed of "1750"? If not: explain how the model is forced between 1750 an 1850.
- Line 117 (Fig. 2): Use "PgC/a" for the units noted in the ordinate labelling (as in

Fig. 3).

- Line 123: Concerning the methodology for determining climate-carbon feedbacks you cite (Zickfeld et al., 2011). Please give credit also to the inventors of this methodology [1, 2].
- Line 147 and Fig. 3: You use the term "changes" with two different meanings: Your "stock changes" are rates of stock change (fluxes), while your "temperature change" refers to a difference. It would be more clear to make this transparent in the text and Figure (as you did in line 164).
- Lines 157-160 (caption Fig. 3): I guess the solid lines show the results for the FOK model, but this is not noted.
- Lines 256 and 282: I do not see how the value of 35% – mentioned as key result of the study in the abstract – follows from the values in table 3. I guess its some kind of mean value, which would not make sense (see next comment).
- Line 234 (Fig. 5): This figure doubles the information from table 3. Moreover, it is inappropriate to show these data as box plots: Box plots are meant to display major characteristics of a statistical distribution, for which it makes sense to calculate mean value and percentiles. But the four scenarios, whose data are listed in table 3, do not form a statistical ensemble so that the results from the scenario simulations are not realizations of a random process which could be characterized by a statistical distribution.

**References**

[1] Cox, Peter M., et al., *Acceleration of global warming due to carbon-cycle feedbacks in a coupled climate model*, Nature 408 (2000) 184-187.

[2] Friedlingstein, P., et al., *How positive is the feedback between climate change and the carbon cycle?*, Tellus B55 (2003) 692-700.

[3] Heinze, Christoph, et al., *ESD Reviews: Climate feedbacks in the Earth system and prospects for their evaluation*, Earth System Dynamics 10 (2019) 379-452.

[4] Hungate, Bruce A., et al., *Nitrogen and climate change*, Science 302 (2003) 1512-1513.

[5] Prentice, I. Colin, et al., *Dynamic global vegetation modeling: quantifying terrestrial ecosystem responses to large-scale environmental change*. In: J.G. Canadell, D.E. Pataki, and L.F. Pitelka (Eds.), *Terrestrial ecosystems in a changing world*, (Springer, 2007), 175-192.

[6] De Sisto, Makcim L., and Andrew H. MacDougall, *Effect of terrestrial nutrient limitation on the estimation of the remaining carbon budget*, Biogeosciences Discussions 2023 (2023) 1-30.

[7] Sulman, Benjamin N., et al., *Multiple models and experiments underscore large uncertainty in soil carbon dynamics*, Biogeochemistry 141 (2018) 109-123.

[8] Todd-Brown, Katherine E.O., et al., *A framework for representing microbial decomposition in coupled climate models*, Biogeochemistry 109 (2012) 19-33.

[9] White, R.E., *Principles and Practice of Soil Science – The soil as a Natural Resource* (Blackwell, Malden, $4^{th}$ ed., 2006)

[10] Zaehle, Sönke, and Daniela Dalmonech, *Carbon–nitrogen interactions on land at global scales: current understanding in modelling climate biosphere feedbacks*, Current Opinion in Environmental Sustainability 3 (2011) 311-320.

---

## Author Comment (AC1)

Carbon-climate feedback higher when assuming Michaelis-Menten kinetics of respiration

Christian Beer

Referee # 1, William Wieder

*Beer uses a reduced complexity model to look at different formulations of a land ecosystem respiration term that uses first order vs. Michaelis-Menten kinetics (FOK and MMK, respectively). The paper reports a stronger land C uptake with the Michaelis-Menten parameterization. The manuscript is a nice, albeit unsurprising, illustration of structural uncertainty in land models and how they impact the magnitude of the terrestrial C sink and potential carbon-cycle climate sensitivities.*

*I like the work, but feel some additional information and clarification is needed to make the findings more straight forward to understand / interpret.*

I would like to thank you for reading carefully this manuscript and for your helpful and constructive questions and suggestions that helped to improve the manuscript.

*Mainly. I'm a little fuzzy about the use of the term carbon-climate feedback here, which seems analogous to the gamma term in the C4MIP literature (e.g. Arora et al 2020; typically expressed as PgC per degree Celsius). With this convention, gamma_land is typically negative (from the atmosphere perspective), reflecting less land carbon storage under warmer conditions.*

*The carbon–climate feedback (γ) quantifies the response of the carbon cycle to changes in physical climate and is expressed in units of carbon uptake or release per unit change in global mean temperature (PgC °C−1).*

*Conceptually, this looks similar to the left side of Fig 1, which focuses on positive feedbacks between temperature, ecosystem respiration and atmospheric CO2 burden. The rest of the results, however, don't share this perspective, which makes the conclusion (and title) confusing.*

Thank you very much for this comment which shows that I need to explain the definition of the metrics used to quantify the carbon-climate feedback in more detail. I will add equations to the text in section 2.3 and hope this increases the clarity. Note, there was a clear mistake in the calculation of the feedback and the feedback factor before as has been noted by the other reviewer.

Confusion can occur because sometimes terms are used in short in the literature. According to the IPCC definition, a climate feedback is an interaction in which a perturbation in one climate quantity (here atmospheric $CO_2$ content) causes a change in a second and the change in the second quantity ultimately leads to an *additional* change in the first due to mechanisms internal to the system. In our case of biogeochemical feedbacks, the feedback is the *additional* change in $CO_2$ after its perturbation due to internal system mechanisms. We start with pre-industrial $CO_2$ content and add anthropogenic $CO_2$ until 2100. Then, we can measure the temporal difference as Delta_CO2. Due to feedbacks, this difference is not pre-industrial $CO_2$ plus the sum of emissions ("$\Delta CO2$ off" in the revised manuscript), but there is a difference which we call feedback f in units of mass of C (Hansen et al., 1984; Lade et al., 2018; Zickfeld et al., 2011) In addition, one can compute the ratio of both temporal CO2

changes, with and without feedback, which is called the feedback factor F (Zickfeld et al., 2011). This factor shows if the feedback is positive (F>1) or negative (F<1) and can be used to compare several biogeochemical feedbacks.

Another question that has been discussing in the literature is the response of land and ocean carbon pools to changes in CO2 or air temperature and assuming a linear relationship $\Delta C_{land} = \beta_{land} * \Delta CO2 + \gamma_{land} * \Delta T$ (Friedlingstein et al., 2003, 2006, Arora et al., 2020). The feedback parameters beta and gamma are sensitivities: How much is land carbon changing due to either CO2 or T effects? Units are mass C per atmospheric concentration or mass C per temperature.

In this paper, I would like to use concentrate on the quantification of the feedback (change in atmospheric CO2) without assuming a linear relationship of the gamma-beta approach. Therefore, the feedback factor is applied. However, because the gamma-beta approach seems to be so manifested in the literature and also is asked for by two reviewers, I have extended the manuscript and added radiatively and biogeochemically coupled simulation experiments in order to estimate these sensitivities following (Friedlingstein et al., 2006; Friedlingstein et al., 2003).

*I'll try to walk though some sources of this confusion and offer suggestions on how to clarify:*

- *Results (Fig 3) shows higher land C uptake and lower atmospheric burden of CO2 with the MMK. To me this implies a reduction in strength of the temperature-respiration feedback with MMK, which allows for more land C uptake. Is this accurate?*

This is a very important question to understand the study. Fig 3 shows changes in state variables of the two model versions when assuming all feedbacks. This is to show how the system of equations work in general and how model results compare to observation-based estimates during the historical period, and how they evolve in future. This shows us the behaviour of the model, e.g. inter-annual variability is not captured, but trends do. The higher land uptake by the MMK model is due to internal carbon dynamics: the CO2 fertilization of GPP leads to more substrate availability for respiration which leads to proportionally more respiration in the case of the linear model (FOK) while the Menten kinetics model assumes a less pronounced impact of more substrate availability and hence reduced respiration relative to the input.

- *The "terrestrial carbon-climate feedback", Table 2, it's unclear if this is basically the size of the cumulative land sink, which is positive (i.e.; from the land perspective) and expressed as Pg C, (not Pg C/degC). Or if this is the difference in atmospheric CO2 accumulation from the "feedback on", (Q10=2) vs. "feedbacks off" (Q10=1). If it's the later, this suggests a larger atmospheric CO2 burden from warming with the MMK approach, which is difficult to square with the results in Fig 3. Maybe the heading for this figure can be clarified?*

I hope that Tab 2 becomes clear with the advanced definition of what is the carbon-climate feedback in section 2.3 of the revised manuscript. Indeed, Tab 2 shows that the feedback is stronger for the MMK model.

- *Finally the "feedback factor" seems to be some kind of a ratio (see additional question below), maybe comparing the runs with Q10=2 ("feedbacks on") vs. Q10=1*

*(feedbacks off)? this may allow diagnoses of the inferred temperatures sensitivity of FOK vs. MMK respiration schemes, but it's not really clear what this metric is communicating, or where the results from the "feedbacks off" simulations come into play here?*

Yes, the feedback factor is explained in detail in (Lade et al., 2018; Zickfeld et al., 2011) but I will extend the methods section using equations to make it more clear here. There is a new definition in the revised manuscript because reviewer #2 identified a mistake. The feedback factor is the ration of two changes in atmospheric CO2 content, including specifically this feedback and without any feedback. Please, have a look in the revised methods section. This dimensionless factor can be used to compare the strength of different feedbacks and different models, and it shows whether the feedback is positive (>1) or negative (<1).

*The simplest solution here may be to remove the use of "carbon-climate feedbacks" here, unless results can be presented in a way that similar to the C4MIP conventions (commonly a reduction in land C uptake per deg warming). If taking this approach, the manuscript can focus on land C sink or land C uptake from the title and throughout the manuscript to be more consistent with results presented. If taking this approach Fig 1 may not be necessary. Clarification on the 'feedback factor' (Table 3, Fig 5) would also be necessary. This may not be accurate, however, if Table 2 is actually showing the difference in atmospheric CO2 burden from Q10=2 vs. Q10=1 experiments.*

I hope to have clarified the term carbon-climate feedback (not the feedback parameter) and how it is calculated, and that I could convince you about its usefulness to understand the uncertainty of model structure for such feedback estimation. In addition, feedback parameters (sensitivities) beta and gamma has been added, too.

Minor and technical questions

*Abstract (and elsewhere?) for a single author paper use "I" not "we".*

Yes I can change the manuscript accordingly.

*I'm not really clear how the "carbon climate feedbacks" were calculated (Table 2), or what the carbon-climate feedback factor represents (Table 3 and Fig 5)? It's the ratio of temporal changes in the land C stocks (end of 21$^{st}$ century pools / end of 19$^{th}$ century pools) Line 134? This doesn't add up when if I do the math on numbers reported in Tables 1 & 2, please clarify in methods. Maybe it's the ratio of the temporal changes in the runs with Q10=2 (feedbacks on) vs. Q10=1 (feedbacks off)? Some addition text in the method and results would help clarify these results.*

See above. Methods section revised.

References

Friedlingstein, P., Cox, P., Betts, R., Bopp, L., Von Bloh, W., Brovkin, V., . . . Zeng, N. (2006). Climate-carbon cycle feedback analysis: Results from the C⁴MIP model intercomparison. *Journal of Climate, 19*(14), 3337-3353.

Friedlingstein, P., Dufresne, J. L., Cox, P. M., & Rayner, P. (2003). How positive is the feedback between climate change and the carbon cycle? *Tellus B: Chemical and Physical Meteorology, 55*(2), 692-700. doi:10.3402/tellusb.v55i2.16765

Hansen, J., Lacis, A., Rind, D., Russell, G., Stone, P., Fung, I., . . . Lerner, J. (1984). Climate Sensitivity: Analysis of Feedback Mechanisms. In *Climate Processes and Climate Sensitivity* (pp. 130-163).

Lade, S. J., Donges, J. F., Fetzer, I., Anderies, J. M., Beer, C., Cornell, S. E., . . . Steffen, W. (2018). Analytically tractable climate--carbon cycle feedbacks under 21st century anthropogenic forcing. *Earth System Dynamics, 9*(2), 507-523.

Zickfeld, K., Eby, M., Matthews, H. D., Schmittner, A., & Weaver, A. J. (2011). Nonlinearity of Carbon Cycle Feedbacks. *Journal of Climate, 24*(16), 4255-4275. doi:https://doi.org/10.1175/2011JCLI3898.1

---

## Author Comment (AC2)

Carbon-climate feedback higher when assuming Michaelis-Menten kinetics of respiration

Christian Beer

Referee # 2

I would like to thank this reviewer for reading this manuscript so carefully, for all the important questions, and for identifying one important misunderstanding of the feedback analysis (see below) which all improved substantially the manuscript.

*I agree to reviewer #1 that, qualitatively, the results from this study are rather foreseeable: Because the MMK model limits soil respiration when the land carbon storage grows beyond a certain amount, this model enhances the land carbon sink, whereby the remaining carbon budget gets larger, the land contribution to the carbon-concentration feedback gets more positive (there is more carbon stored per ppm CO2 rise; _ sensitivity), and the land contribution to the climate-carbon feedback gets more negative (there is more soil carbon to be respired per degree climate change; sensitivity). So the main advancement by this study would be its quantitative results (that are partly noted in the abstract). But I doubt that those numbers have any relevance (major comment 1 below) and, moreover, I don't trust these numbers (major comment 2 below). In addition, as also noted by reviewer #1, the feedback analysis is rather unclear (major comment 3 below), and – if my reconstruction of its meaning is right – the feedback metrics used are not simply a measure of the carbon-climate feedback (as intended by the author; see title) but are mixing information on this and the carbon-concentration feedback.*

I agree with the reviewer that it is important to quantify the effect of a Menten kinetics formulation for the carbon-climate feedback. That is actually the aim of the study and the methods used. The numbers presented shows the order of magnitude of the effect of the different respiration formulation itself. This shows the strong need to advance our theoretical understanding of which assumptions and equations to be used at which scale. I will clarify all major and minor points raised by the reviewer below.

This study aims to quantify the effect of the respiration model formulation used on the response of the carbon cycle and climate to anthropogenic emissions and hence the carbon-climate feedback. This is an important question because different model formulations are used in the literature and parameters calibrated based on recent observations. Here, I show that different formulations of the respiration differential equation lead to different responses to emissions even when the models are fitted to recent observations.

Major 1)

I agree with the reviewer that the coupling of carbon and nutrient cycles can largely influence the response of ecosystems to climate change and hence climate feedbacks. This is also true for many other interactions within ecosystems which are not considered in this study, e.g. the coupling to the hydrological cycle: soil carbon stocks influence soil water content and hence evapotranspiration and also GPP and plant growth. This study aims to quantify the effect of the respiration model formulation used on the response of the carbon cycle and climate to anthropogenic emissions and hence on the carbon-climate feedback. For this, a simplified 0-dimentional model is used because the overall order of magnitude of the effect is of interest. To include many other important effects on the carbon cycle is interesting but out of the scope of this study, and could be still hard to be done even using a most advanced Earth System Model. For example, coupling of carbon, water and nutrient cycles and using a Menten

kinetics formulation of heterotrophic respiration is represented in (Yu et al., 2020) but global simulations are still work in progress and I do not know any respective experiments using a full coupled ESM simulation.

Still, I fully agree with the reviewer that these additional ecosystem interactions are generally important for a quantification of the climate-carbon feedback and will advance the discussion section of the revised manuscript in this respect.

On the usage of either equation for respiration: This is more a scaling question than a question of biogeochemical cycles interaction. The underlying biogeochemical reactions are all enzymatic, hence at the process level we should apply a Menten kinetics model with some assumptions like the transition complex being in steady state. At the macro scale, the same model can be used when all state variables and environmental conditions are correctly integrated (Reichstein & Beer, 2008). This is for example how we operate in land surface models for estimating photosynthesis: The Farquhar model is a Menten kinetics model and we use it at the landscape scale integrating the leaf area index. However, sometimes we can simplify the Menten kinetics to a first-order model, e.g. when the amount of substrate is much smaller than the Menten parameter K. Such first-order model is the traditional model used (Andrén & Kätterer, 1997; Parton et al., 1988) in ESMs but novel developments assume Menten kinetics now (Yu et al., 2020) This led to the research question of this paper: what is the structural uncertainty of this model formulation on future carbon cycle functions and the carbon-climate feedback even when parameters are calibrated such that recent fluxes are similar to observations? I will make sure in the revised manuscript introduction section that this rational is more clear.

Major 2)

a) Land carbon cycle model formulation. Thank you very much for thinking in depth into this problem.

There is a misunderstanding about the respiration term in the model: This is not heterotrophic respiration but ecosystem respiration as the sum of autotrophic and heterotrophic respiration. This is clear from the second term of the equation in (Lade et al., 2018) which multiplies the decomposition rate constant with the total land carbon pool. Indeed, I was co-author of the study (Lade et al., 2018) in particular to oversee the land carbon model formulation.

For this study, I recognized this mistake in the previous formulation and made the model consistent. The first term of the equation represents GPP ($CO_2$ fertilization works on GPP not NPP), the second term ecosystem respiration and the last term land use emissions. That is also why the parameters needed to be adjusted in the FOK model version: the decomposition rate constant k is the ratio of GPP to total land carbon following (Carvalhais et al., 2014). Fig 2 shows that this more consistent model also compares well to observations.

b) MMK model parameter estimation. I adjusted the two Menten parameters vmax and K have in order to match historical observations and FOK model results in the previous version of the manuscript. I agree with the reviewer that this calibration could have been done in a more formal way. For the revised version of the manuscript, I used a formal least-squares gradient decent method (MATLAB function lsqnonlin) to calibrate the two Menten kinetics parameters against observations of the landatmosphere exchange of CO2 for the historial period. The such slightly adjusted parameters are vmax=200 PgC a$^{-1}$ and K=1787 PgC. MMK model results are similar to the previous version. All results will be re-computed using these parameters and the methods section extended.

Major 3)

I would like to thank the reviewer to think about the feedback analysis in that detail and for pointing out an important previous misconception on my side. I fully agree with the reviewer to estimate the carbon-climate feedback as the difference in CO2 change of two simulations: one switching off all feedbacks ("off") and one having considered the feedback of interest, here the carbon-climate feedback ("on") following the procedure explained in (Hansen et al., 1984; Zickfeld et al., 2011). With that the feedback factor is defined as F=ΔCO2(on)/ΔCO2(off). This way the carbon-concentration feedback does not influence the analysis of the carbon-climate feedback. The methods section is extended in order to explain the procedure and calculations in detail.

I can understand that the audience wants to compare also feedback parameters (sensitivities) beta and gamma following (Friedlingstein et al., 2006; Friedlingstein et al., 2003) and I present these numbers in the revised manuscript and extend the methods section accordingly.

After applying the adjusted concept, the feedback factors are similar and the overall conclusions do not change (table below). Beta and gamma values of the FOK model simulation assuming the high-emission pathway (SSP5-85) are at the higher end of the range of values reported in the literature (Arora et al., 2020; Friedlingstein et al., 2006; Zickfeld et al., 2011). Beta and gamma (and feedback factors) increase towards low-emission scenarios (REF) and also amplify when using the Menten kinetics model MMK (table below). I will extend the discussion section with these comparisons.

| Feedback factor | First-order kinetics (FOK) | Michaelis-Menten kinetics (MMK) |
|---|---|---|
| SSP1-26 | 1.41 | 2.0 |
| SSP2-45 | 1.31 | 1.71 |
| SSP3-70 | 1.23 | 1.52 |
| SSP5-85 | 1.21 | 1.43 |

| Feedback parameters | β, PgC ppm$^{-1}$, first-order kinetics (FOK) | β, PgC ppm$^{-1}$, Michaelis-Menten kinetics (MMK) | γ, PgC K$^{-1}$, first-order kinetics (FOK) | γ, PgC K$^{-1}$, Michaelis-Menten kinetics (MMK) |
|---|---|---|---|---|
| SSP1-26 | 3.4 | 9.8 | -133 | -218 |
| SSP2-45 | 2.3 | 5.4 | -125 | -204 |
| SSP3-70 | 1.6 | 3.4 | -124 | -198 |
| SSP5-85 | 1.4 | 2.7 | -117 | -187 |

Minor

*Title, abstract, and throughout: Whenever the author talks of respiration, heterotrophic respiration is meant (also called "soil respiration"), in contrast to autotrophic rspiration. This should be made clear in the title, the abstract, and elsewhere.*

As explained above, the model considers ecosystem respiration.

*The Lade et al. model is missing the temperature dependence of photosynthetic production. I don't think that a quantification of the carbon-climate feedback without that temperature dependence makes sense. At least it should be made clear that the study accounts (if at all) only for part of that feedback.*

On a global scale, there is no clear relation of GPP to temperature; individual plant functional types can have different low and high temperature inhibition functions (Sitch et al., 2003), but in general photosynthesis is most constrained by light, CO2 and nutrients. The indirect effect of temperature on soil water content and hence stomatal conductivity, and the limitation of photosynthesis by light and CO2 will override any specific increase of photosynthesis with temperature. But also at the individuum scale, there is no clear relationship, see Fig 9.16 in https://www.ehleringer.net/uploads/3/1/8/3/31835701/413.pdf Temperature dependence of autotrophic respiration is accounted for by the model as full ecosystem respiration is considered.

*Line 9 (abstract): The formulation "The epistemic uncertainty . . . is unclear" a bit weird: I guess the author wants to express that because of epistemic uncertainty there is quantitative uncertainty whose size is unclear – hence it would be the quantitative uncertainty being unclear, not the epistemic uncertainty.*

I agree and update this sentence to: "The effect of the respective mathematical representations on the terrestrial carbon-climate feedback is unclear."

*Lines 13-15 (abstract): This formulation is misleading: taking it literally, the author suggests to improve our understanding of the model structure of Earth system models. But I guess the author wants to emphasize that we need to improve our understanding of heterotrophic respiration to improve its representation in Earth system models.*

No, it is less the understanding at the process level that is missing but more the question on how to model these processes at a landscape to global scale.

*Line 28: Why "in contrast"? In contrast to what?*

Words deleted. Thanks.

*Line 35 (caption of Fig. 1) and line 39: Only the negative feedback may be termed "biogeochemical", the positive feedback is non-biological and mostly determined by physics (radiation in the atmosphere, temperature dependenc of reaction rates); for terminolgy see e.g. [3].*

Following IPCC AR6, chapter 5, we define biogeochemical feedbacks as feedbacks that change atmospheric greenhouse gas concentrations due to biological functions which is the case for both the carbon-concentration feedback (biological function photosynthesis) and the carbon-climate feedback (biological respiration processes), cf. Fig 1. See also (Arneth et al.,

2010). In contrast, a change in vegetation cover hence albedo change as a response to CO2 and climate would form a biogeophysical feedback mechanism.

*Line 50: A better reference than (Zickfeld et al., 2011) on the methodology of separating climate-carbon feedbacks would be the original study [2] or the recent review [3].*

Thank you for pointing on these references. For the original methodology of the feedback factor I will include also Hansen et al (1984). This is included into the revised version of the manuscript.

*Line 85: Better: "atmospheric CO2 concentration" instead of "atmospheric carbon content".*

The model operates with the latter one, units in PgC, see methods section parameter table. But I will extend the methods section to make this more clear.

*Line 106: Should this read "1850" instaed of "1750"? If not: explain how the model is forced between 1750 an 1850.*

We run the model during the time period 1850-2100. The model is initialized with values at 1850 and driven by emissions. Land-use change emissions were available since 1850 from the Global Carbon Project.

*Line 117 (Fig. 2): Use "PgC/a" for the units noted in the ordinate labelling (as in Fig. 3).*

I harmonize the text using PgC a$^{-1}$

*Line 123: Concerning the methodology for determining climate-carbon feedbacks you cite (Zickfeld et al., 2011). Please give credit also to the inventors of this methodology [1, 2].*

Done.

*Line 147 and Fig. 3: You use the term "changes" with two different meanings: Your "stock changes" are rates of stock change (fluxes), while your "temperature change" refers to a difference. It would be more clear to make this transparent in the text and Figure (as you did in line 164).*

Carbon fluxes are reaction rates calculated as stock change per year. Fig 3 also shows the temperature anomaly, here the difference to the pre-industrial average is meant. I will use terms "carbon flux" and "temperature anomaly" for more clarity in the revised version of the manuscript.

*Lines 157-160 (caption Fig. 3): I guess the solid lines show the results for the FOK model, but this is not noted.*

Thanks, added to the caption.

*Lines 256 and 282: I do not see how the value of 35% – mentioned as key result of the study in the abstract – follows from the values in table 3. I guess its some kind of*

*mean value, which would not make sense (see next comment).*

*Line 234 (Fig. 5): This figure doubles the information from table 3. Moreover, it is inappropriate to show these data as box plots: Box plots are meant to display major characteristics of a statistical distribution, for which it makes sense to calculate mean value and percentiles. But the four scenarios, whose data are listed in table 3, do not form a statistical ensemble so that the results from the scenario simulations are not realizations of a random process which could be characterized by a statistical distribution.*

Thank you for this clarification. I remove the boxplot, and discuss the four individual relative differences in the revised text.

References

Andrén, O., & Kätterer, T. (1997). ICBM: THE INTRODUCTORY CARBON BALANCE MODEL FOR EXPLORATION OF SOIL CARBON BALANCES. *Ecological Applications, 7*(4), 1226-1236.

Arneth, A., Harrison, S. P., Zaehle, S., Tsigaridis, K., Menon, S., Bartlein, P. J., . . . Vesala, T. (2010). Terrestrial biogeochemical feedbacks in the climate system. *Nature Geoscience, 3*(8), 525-532. doi:10.1038/ngeo905

Arora, V. K., Katavouta, A., Williams, R. G., Jones, C. D., Brovkin, V., Friedlingstein, P., . . . Ziehn, T. (2020). Carbon–concentration and carbon–climate feedbacks in CMIP6 models and their comparison to CMIP5 models. *Biogeosciences, 17*(16), 4173-4222. doi:10.5194/bg-17-4173-2020

Carvalhais, N., Forkel, M., Khomik, M., Bellarby, J., Jung, M., Migliavacca, M., . . . Reichstein, M. (2014). Global covariation of carbon turnover times with climate in terrestrial ecosystems. *Nature, 514*(7521), 213-217.

Friedlingstein, P., Cox, P., Betts, R., Bopp, L., Von Bloh, W., Brovkin, V., . . . Zeng, N. (2006). Climate-carbon cycle feedback analysis: Results from the C⁴MIP model intercomparison. *Journal of Climate, 19*(14), 3337-3353.

Friedlingstein, P., Dufresne, J. L., Cox, P. M., & Rayner, P. (2003). How positive is the feedback between climate change and the carbon cycle? *Tellus B: Chemical and Physical Meteorology, 55*(2), 692-700. doi:10.3402/tellusb.v55i2.16765

Hansen, J., Lacis, A., Rind, D., Russell, G., Stone, P., Fung, I., . . . Lerner, J. (1984). Climate Sensitivity: Analysis of Feedback Mechanisms. In *Climate Processes and Climate Sensitivity* (pp. 130-163).

Lade, S. J., Donges, J. F., Fetzer, I., Anderies, J. M., Beer, C., Cornell, S. E., . . . Steffen, W. (2018). Analytically tractable climate--carbon cycle feedbacks under 21st century anthropogenic forcing. *Earth System Dynamics, 9*(2), 507-523.

Parton, W. J., Stewart, J. W. B., & Cole, C. V. (1988). Dynamics of C, N, P and S in grassland soils: a model. *Biogeochemistry, 5*, 109-131.

Reichstein, M., & Beer, C. (2008). Soil respiration across scales: the importance of a model--data integration framework for data interpretation. *Journal of Plant Nutrition and Soil Science, 171*(3), 344-354.

Sitch, S., Smith, B., Prentice, I. C., Arneth, A., Bondeau, A., Cramer, W., . . . others. (2003). Evaluation of ecosystem dynamics, plant geography and terrestrial carbon cycling in the LPJ dynamic global vegetation model. *Global Change Biology, 9*(2), 161-185.

Yu, L., Ahrens, B., Wutzler, T., Schrumpf, M., & Zaehle, S. (2020). Jena Soil Model (JSM v1.0; revision 1934): a microbial soil organic carbon model integrated with nitrogen and phosphorus processes. *Geoscientific Model Development, 13*(2), 783-803. doi:10.5194/gmd-13-783-2020

Zickfeld, K., Eby, M., Matthews, H. D., Schmittner, A., & Weaver, A. J. (2011). Nonlinearity of Carbon
    Cycle Feedbacks. *Journal of Climate, 24*(16), 4255-4275.
    doi:https://doi.org/10.1175/2011JCLI3898.1

---

## Author Response (AR2)

Dear Author,

Thank you for your revised manuscript submission to ESD. The revised manuscript has been assessed by two reviewers. They note the potential value of the analysis, but have also requested major revisions; these include further clarification on aspects of the analysis, and changes in the presentation of your results and conclusions. Please address all their comments in your response, especially the reviewer comments on (i) the suitability of a simple 0-D model to provide quantitative analysis of the carbon budget, (ii) the request for further uncertainty analyses to strengthen your conclusions.

Reviewer 3

The study addresses a question of differing model-structures and their implications on forecasts of the global carbon cycle. The challenge of quantifying and resolving structural uncertainty has been largely ignored, with the implicit assumption of adding more detail makes a model better, which often not true. So on that basis I welcome this study and its results as they could provide an easy way of directing attention onto an important issue as a 'call to arms'.

I like to thank the reviewer for a carefully reading the manuscript and for a detailed review. This discussion is very useful and led to a further improvement of the manuscript. I will answer in detail below and track changes in a separate document.

However, the manuscript as written leaves many appropriate caveats which sets the context of the study to the final paragraph of the discussion. That leaves the introduction with the narrative of the Michaelis-Menten approach being more realistic (e.g. L57), but no empirical evidence of this at the scale of the model is given. This is also implied by the title of the article.

Thank you very much for this comment. Indeed, we usually discuss from the point of view of the process level, that MMK should be the more valid approach because the underlying biochemical reactions are enzymatic. However, both FOK and MMK come with assumptions at the landscape scale and hence it remains unclear, which approach is more valid at this large scale. I have addressed this very clearly in the discussion section, but I see that I used unbalanced sentences in the introduction section. Therefore, I rewrote the introduction section accordingly. See LThe idea of the paper is not to demonstrate that one of the approaches is more suited but to show the related structural uncertainty.

Similarly, the result and discussion are written from the perspective of differences between approaches as themselves of value rather than the size of difference in context of the GCB.

This is a very valid point. The idea of this paper is not to per se quantify carbon budgets or feedbacks. For this, an ESM is the valid tool. I only want to address the structural uncertainty that we face in the current models, when using either of the two approaches. Still, to demonstrate this uncertainty we need to see quantitative differences between the approaches and discuss them (see also discussion round 1 with reviewer #2). Please, see changes in abstract, discussion, and conclusion sections in which I focus more on the qualitative difference now.

Think the study would be best used to quantification of how large structural uncertainty can be and should be presented more in that light, e.g. that the structural uncertainty of this 1 component is potentially equal to a nearly years GPP, or greater than 1 years heterotrophic respiration.

I agree and compared the difference in the remaining C budgets for example with projected C release from permafrost ecosystems or other underrepresented feedbacks. I advance this discussion by also comparing the ca 100 PgC difference in the remaining C budget under SSP2-45 with a one year GPP or respiration flux (130 PgC), see L296.

Some minor comments on the text:
1) The calibration of both models to the same contemporary observations should be clearer. Some quantification of their ability to fit the contemporary period would also be interesting. In Figure 3 the results of the MMK appear to be diverging, to what extent these are still consistent with independent estimates would be useful.

This topic has been also discussed in the first round of reviews and led to a revised calibration of the MMK model parameters. As written in section 2.1, the FOK parameters are usually taken from Lade et al. (2018) but adjusted accordingly because I improved the terrestrial C equation. The tuning parameters lambda and alpha have been adjusted in order to represent pre-industrial C budgets and temperature trends. However, the focus of the paper is a comparison of FOK and MMK model results, there is no need to exactly match historical observation-based estimates (interannual variability is anyhow not possible). What is important here, is to find MMK parameters for which both models show similar pre-industrial results because we want to analyse feedback differences. For this, I explained in section 2.1 that I used a gradient decent method to optimize MMK parameters accordingly, because these parameters are most unclear and could not taken from literature. After 1950 we see the climate change signal and hence already FOK and MMK deviations.

2) That the study is mixing autotrophic and heterotrophic respiration, even though their dependences and links to total C are different. This again links back to the need to make clear that the results are illustrative not to be interpreted.

I agree, the model is simplified such that both respiration components are lumped together into one ecosystem respiration process. Related limitations are clearly discussed in L301 and following. I add here this point of only looking at one total respiration in L302.

3) The implicit assumptions of the two models could be clearer. FOK assumes that microbial pool will continue to grow thus maintaining a decomposition, where MMK assumes that the microbial pool remains unchanged. Both of these extremes are probably unrealistic.

Thank you very much for this hint, which improves the introduction section. See new L57-58 and L63.

4) Consider introducing more of the information from L292-311 earlier in the manuscript to support the narrative of structural uncertainty is important but that this is illustrative.

I would like to avoid repeating discussion section content in the introduction section, but I agree with the reviewer that we should have a hint on the model limitations upfront, see new lines L84-87 in the introduction section.

Reviewer 4

Beer presents an analysis of the carbon-climate feedback using two different formulations (first order kinetics, FOK, vs Michaelis-Menten kinetics, MMK) for calculating the terrestrial ecosystem respiration in a simple simple analytical global carbon cycle - climate model.

The value in this study lies in the qualitative understanding of the resulting differences in the feedback factor when using the different mathematical formulations. However, Beer presents a quantitative analysis suggesting that the terrestrial carbon-climate feedback doubles when assuming Michaelis-Menten kinetics. He concludes that the remaining carbon budget to keep global warming below 2°C is 66-113 Pg C higher when using MMK compared to FOK.

I would like to thank this reviewer for reading the manuscript in detail and for all the valuable comments which helped to further improve the paper. Thanks for seeing the value of the study. The idea is indeed to show qualitative effects and do not interpret exact numbers, hence orders of magnitudes are discussed in the paper. We also need to calculate some numbers in order to give such qualitative conclusions – that is for what the simple model is about. I went through abstract, discussion and conclusions again to make this more clear.

These figures are derived from a simple, 0-dimensional model and are presented without any uncertainty analysis. My first concern is whether such a simple model is suitable to perform a quantitive analysis of the remaining carbon budget when using a different respiration formulation.

Of course, the model is most simplified and we should not interpret concrete single numbers. The idea is indeed to see the order of magnitude differences when assuming FOK vs MMK. However, in general, and despite its simplicity, the model is able to capture global trends in historical atmosphere, land and ocean C dynamics, and the global temperature trend. In addition, projections are similar to published IPCC model results with some deviation, e.g. an overestimation of the future ocean C sink. Compare L158-183. The model has been designed to study feedbacks (Lade et al., 2018).

And the second concern is that, since the analysis is based on a simplistic model, no uncertainty analysis of the estimated feedback and carbon budgets is performed.

Model projections and related feedback or remaining C budget estimations come with different kinds of uncertainty. This paper concentrates on the structural model uncertainty, that is the uncertainty related to different formulations of the equation for one process (here, respiration). I can show that this uncertainty is huge, feedback estimates can even double.

In addition, we can express uncertainty related to the parameters used in the model. This was not the focus of the paper, but indeed it can be interesting to compare such different kinds of uncertainty. Therefore, I selected two most sensitive parameters for the carbon-climate feedback (lambda and Q, Tab. 1), resampled 100 pairs of both using Latin Hypercube Sampling and assuming a standard deviation of 0.4 for lambda and 0.2 for Q in order to derive a reasonable parameter range (lamda=[1.5,3.5] and Q=[1.6,2.6]), and repeated the feedback and feedback factor calculation and the respective model runs using these 100 samples of parameters. The results are shown in the two new boxplots below. We can see a clear parameter uncertainty, but the structural uncertainty of the formulation of the respiration equation is even higher. Mean values between FOK and MMK result distributions are significantly different. That means that despite the parameter uncertainty, the conclusions about the structural uncertainty hold.

[Figure]

**Carbon-climate feedback (PgC)**

[Figure]

**Carbon-climate feedback factor (-)**

---

## Author Response (AR3)

**Carbon-climate feedback higher when assuming Michaelis-Menten kinetics of respiration**

Christian Beer[1, 2]

[1]Department of Earth System Sciences, Faculty of Mathematics, Informatics, and Natural Sciences, University of Hamburg, Hamburg, 20134, Germany
[2]Center for Earth System Research and Sustainability, University of Hamburg, Hamburg, 20134, Germany

*Correspondence to*: Christian Beer (christian.beer@uni-hamburg.de)

Reply to editor comments, round 3

19 June 2025

Dear Editor

Thank you for going through the revised manuscript and the replies to the reviewer comments in detail. I have added the parameter uncertainty analysis as additional supplemental information to the manuscript and included the conclusions from that analysis into the discussion section of the manuscript. Further, spelling mistakes are corrected and the referencing of literature adjusted. In addition, the model code and all code to redo the plots has been published at zenodo and the doi is provided.

Yours Sincerely

Christian Beer